# The Impact of Sex, Body Mass Index, Age, Exercise Type and Exercise Duration on Interstitial Glucose Levels during Exercise

**DOI:** 10.3390/s23229059

**Published:** 2023-11-09

**Authors:** Ninoschka C. D’Souza, Durmalouk Kesibi, Christopher Yeung, Dorsa Shakeri, Ashwin I. D’Souza, Alison K. Macpherson, Michael C. Riddell

**Affiliations:** 1School of Kinesiology and Health Science, Faculty of Health, York University, Toronto, ON M3J 1P3, Canada; noshd5@yorku.ca (N.C.D.); durrak@my.yorku.ca (D.K.); chrisyeung04@gmail.com (C.Y.); sdorsa@yorku.ca (D.S.); alison3@yorku.ca (A.K.M.); 2The Hospital for Sick Children, Toronto, ON M5G 0A4, Canada

**Keywords:** continuous glucose monitoring (CGM), exercise, hypoglycemia, hyperglycemia, BMI, age

## Abstract

The impact of age, sex and body mass index on interstitial glucose levels as measured via continuous glucose monitoring (CGM) during exercise in the healthy population is largely unexplored. We conducted a multivariable generalized estimating equation (GEE) analysis on CGM data (Dexcom G6, 10 days) collected from 119 healthy exercising individuals using CGM with the following specified covariates: age; sex; BMI; exercise type and duration. Females had lower postexercise glycemia as compared with males (92 ± 18 vs. 100 ± 20 mg/dL, *p* = 0.04) and a greater change in glycemia during exercise from pre- to postexercise (*p* = 0.001) or from pre-exercise to glucose nadir during exercise (*p* = 0.009). Younger individuals (i.e., <20 yrs) had higher glucose during exercise as compared with all other age groups (all *p* < 0.05) and less CGM data in the hypoglycemic range (<70 mg/dL) as compared with those aged 20–39 yrs (*p* < 0.05). Those who were underweight, based on body mass index (BMI: <18.5 kg/m^2^), had higher pre-exercise glycemia than the healthy BMI group (104 ± 20 vs. 97 ± 17 mg/dL, *p* = 0.02) but similar glucose levels after exercise. Resistance exercise was associated with less of a drop in glycemia as compared with aerobic or mixed forms of exercise (*p* = 0.008) and resulted in a lower percent of time in the hypoglycemic (*p* = 0.04) or hyperglycemic (glucose > 140 mg/dL) (*p* = 0.03) ranges. In summary, various factors such as age, sex and exercise type appear to have subtle but potentially important influence on CGM measurements during exercise in healthy individuals.

## 1. Introduction

Exercise increases liver glucose production and muscle glucose uptake by ~five- to sevenfold, which can disrupt glucose homeostasis, even in people not living with diabetes [1]. The interstitial fluid, which supplies the cells of the body with glucose, typically provides only a small reservoir of circulating glucose, about 15 g in the average adult who weighs ~70 kg, which is in rough equilibrium with the ~5 g of glucose in the bloodstream [1,2,3]. In men [1] and women [4] without diabetes who are exercising while fasted in a laboratory setting, the rate of glucose appearance into the bloodstream largely matches the rate of disappearance during continuous moderate-intensity aerobic exercise, with venous (or capillary) blood glucose levels typically measured between 80 and 100 mg/dL. However, exposure to low blood glucose concentrations (i.e., circulating levels < 70 mg/dL) can occur during routine exercise, particularly if a rise in insulin secretion is triggered via simple carbohydrate feeding before exercise [5,6], or if the exercise duration is prolonged (i.e., >1–2 h) without carbohydrate feeding [7,8,9]. In contrast, an elevation in venous blood glucose concentration well above the so-called “normal range” for glucose in healthy individuals (healthy can be defined as a venous glucose measurement of 70–140 mg/dL) can be observed if there is carbohydrate loading before and/or during the activity [10,11] or if the exercise is particularly brief and intense [12]. Some small sex-specific differences have been reported for glucose turnover and glycemia during exercise [4,13], but sex-related differences have not always been found when looking at capillary or venous glucose levels intermittently in laboratory-based studies [14,15,16]. The small number of subjects in these laboratory studies, and the infrequent glucose monitoring, likely limit their ability to detect potentially small sex-related differences in blood glucose levels, if they do exist. The impact of age, body mass index (BMI) and exercise type on whole blood, plasma or interstitial glucose levels during exercise in the apparently healthy population is also unclear but has been explored in individuals living with type 1 diabetes where sex- and BMI-specific differences have been reported at least for interstitial glucose measurements [17].

With the recent uptake of frequent real-time continuous glucose monitors (CGMs) in the general population [18], and with new glucose sensor technologies in constant development, larger free-living studies are needed to better understand the “normative” CGM data trends for exercise in men and women who are not living with diabetes. CGM is a newer wearable technology that measures interstitial glucose concentrations approximately every 1–5 min by using a glucose oxidase sensor filament inserted under the skin, into the interstitial fluid, which is connected to a transmitter placed on the body above the sensor using an adhesive [19]. Interstitial glucose values are collected and stored via communication between the sensor implanted under the skin and the transmitter that sits on top of the sensor and a peripheral receiver, typically a smart phone with a CGM application. In general, CGM values between 70 and 140 mg/dL (3.9–7.8 mmol/L) are considered “within the *normal* or *tight* glycemic range” in healthy individuals, depending on if the individual is fasted or not, with values above and below this range considered to indicate “high” or “low” glucose, respectively [20]. The CGM data distribution within, below and above this so-called normal glycemic range during exercise in people without diabetes is less understood, but CGM data at rest [21,22,23,24,25] and during limited forms of exercise [22,26,27] have been published from healthy populations. Unfortunately, CGM accuracy overall during exercise may be less than what is observed at rest, at least in people living with type 1 diabetes [28], but it is likely that any sensor error would be consistent when generating a normative dataset that compares user characteristics, such as age, sex and/or body mass index status. The utility of CGM as a performance-enhancing tool or a monitoring tool for physically active people, including athletes without diabetes, has more recently been described [29,30], but again a normative dataset might be of some comparative value given the expanse of CGM use in the nondiabetic population, who tend to range widely in age, weight and exercise preferences.

The International Hypoglycemia Study Group, who focus primarily on the use of CGM and other metrics in diabetes, have recently stated that a threshold of blood or interstitial glucose of <54 mg/dL (<3.0 mmol/L), which is thought to be rare in people who are not on insulin therapy, or on an insulin secretagogue, should be used when reporting on Level 1 “clinically important hypoglycemia” (i.e., impairs cognition, promotes cardiac arrhythmia, etc.), while values >53 mg/dL but <70 mg/dL are to be considered Level 1 hypoglycemia “alerts” [31]. Since healthy individuals may have CGM values above or below the so-called “target” range of 70–140 mg/dL, in settings of prolonged fasting, endurance exercise or “excessive” feeding [20,25,26,32], new criteria should be set for the use of CGM in individuals who are not living with diagnosed dysglycemia (i.e., diabetes, prediabetes). CGM might even hold promise in helping detect early stages of dysglycemia and/or hypoglycemia. While clearly valuable in diabetes populations, CGM use in the nondiabetic marketplace has been questioned recently [18], primarily because clinical trials have yet to show any value from device use and because it is generally held that glucose homeostasis is typically not impacted much by food or exercise in people not living with a diagnosis of diabetes. In one study of normal glucose tolerance adults, the percent of time with CGM values < 70 mg/dL at rest was small overall, but statistically higher in women (3.2%) as compared with men (1.7%), at least according to one first-generation CGM device [33]. In another more recent study, sex was not shown to be a factor in the change in interstitial glucose levels caused by exercise or meals, but the distribution of CGM data was not reported [26]. This study is a secondary analysis of a previously published dataset [25,26], using a new analysis, where 119 apparently healthy individuals wore blinded CGMs during their usual exercise activities for up to 10 days. The primary purpose of this analysis was to profile the percent of time in various predefined glucose ranges during different forms of exercise (i.e., predominantly aerobic, resistance or mixed) in apparently healthy adult men and women. A secondary analysis included using generalized estimating equation (GEE) models to test the hypotheses that sex, age and body mass index (BMI) influence the distribution of glucose levels in these predefined zones. These data will aid in establishing set normative ranges for individuals without diabetes while factoring in covariates of age, sex and exercise type and duration. It is our intent to establish these normative ranges for further application in future studies involving glycemic management with and without diabetes.

## 2. Methods

### 2.1. Study Design and Participants

This study was conducted as a secondary analysis from an original dataset that collected CGM data around meals and exercise in nondiabetic individuals [25,26]. In brief, study participants were apparently healthy individuals and not living with dysglycemia or diabetes, as verified by a point-of-care HbA1c level at study entry of <5.7% (39 mmol/mol). They were recruited from family, friends and neighbors of patients living with a diagnosis of type 1 diabetes (T1D) who were seen in one of 12 major US diabetes clinics (T1D Exchange Clinic Network). Major eligibility criteria were age ≥6 years; body mass index <30.0 kg/m^2^ for participants ≥18 years old or BMI percentile between the 5th and 85th percentile for participants <18 years old; no chronic illness or medications that might affect glucose metabolism; and point-of-care HbA1c <5.7% (39 mmol/mol). Female participants pregnant at the time of study enrollment were not eligible. All exercise events were performed at home or in a community setting, as previously described [26], with each participant completing at least one or more separate exercise sessions over a 10-day period. Interstitial glucose data for all participants were recorded using a blinded CGM (Dexcom G6) with the user’s activity recorded, using an activity log, to identify the exercise start time and end time to the nearest minute. Exercise modalities were categorized by the users using a diary as predominantly aerobic, predominantly resistance or mixed (both aerobic and resistance exercise), which is similar to how at-home exercise is categorized in other CGM-based studies for statistical purposes [34]. Exercise data were included only if the exercise duration was 10 min or more, since the CGM system used in this study only provides 5 min filtered values for interstitial glucose. The participant study diary was used to identify activity events that were then categorized as aerobic, resistance or mixed. Only exercise sessions that occurred at least 30 min after a meal were included in the analysis to help reduce the confounding influence of meals on exercise-associated glycemia. Time of day for all sessions was between 5 am and 12 am midnight, but the time of day was not considered as a potential CGM-influencing factor. For each exercise event, the baseline (i.e., pre-exercise) glucose, mean glucose, nadir glucose, peak glucose and postexercise glucose concentrations were determined. CGM data during exercise were also categorized into glycemic zones for each exercise activity: <54 mg/dL (very low); <70 mg/dL (low); 70–120 mg/dL (normal); >120 mg/dL (elevated); >130 mg/dL (high); >140 mg/dL (extremely high). Three or more CGM values in a row <70 mg/dL were documented as sustained biochemical hypoglycemia [35], whereas values <54 mg/dL were deemed as clinically significant hypoglycemia [31]. For these analyses, predefined comparisons were made for the above-mentioned CGM metrics (e.g., time in tight range, change in glucose, etc.) for the following categorical variables serving as covariates: sex, age, body mass index status and exercise duration only (see statistical analyses section below).

This study was approved by the institutional review boards at 12 US diabetes centers within the Type 1 Diabetes Exchange Clinic Registry [25,26]. Written informed consent was obtained from adult participants (age ≥ 18 years) and parents/guardians of minor participants (age < 18 years). Minor participants also provided written assent, according to local institutional review board requirements.

### 2.2. Statistical Analysis

Statistical analysis was carried out using a repeated measures approach with generalized estimating equation (GEE) models, to account for multiple exercise events completed by each participant that may have differed in exercise type and duration. This analysis was carried out on IBM SPSS Statistics 29 Software. GEE adds repeated measures capabilities to a generalized linear model [36]. It is an approach that accounts for the correlation between the repeated measures of each subject to increase efficiency and gives a consistent population average parameter and variances under weak assumptions [36]. In this study, exercise type and duration were repeatedly measured over time for all subjects, thus creating clusters. In the GEE model, linear regression was selected. The dependent variables included interstitial glucose levels measured at baseline (i.e., pre-exercise), postexercise glucose, average glucose level during exercise, nadir glucose level during exercise and peak glucose level during exercise. Covariates included in the model were sex (male/female), BMI (underweight, normal BMI, overweight), exercise type (aerobic, resistance and mixed), exercise duration and age. The GEE model was set up to control for differences created by the covariates. The pre- and postexercise glucose values were extracted from the participant’s CGM data and diaries using the 5 min CGM averages for the closest aligned time point. A final GEE analysis of the difference between peak and nadir glucose for each subject at each session was also assessed. Analyses of percent of time in various interstitial glucose ranges were also performed. For this, we predefined the interstitial glucose level to be in the “tight glycemic range” during exercise if the CGM value was 70–120 mg/dL, below target range if the value was <70 mg/dL, slightly above target range if the value was >120, ≤130 mg/dL, elevated if the value was >130, ≤140 mg/dL or hyperglycemic if the value was >140 mg/dL. Biochemical hypoglycemia was defined as 15 consecutive minutes, or more, with CGM values <70 mg/dL, as per recent CGM consensus guidelines for people living with diabetes [37]. This criterion may be overly conservative for individuals not living with diabetes, given they would have intact glucose counterregulation and minimal exposure time to biochemical hypoglycemia, unless repeated prolonged bouts of exercise are performed over subsequent days [38], so we also conducted analyses on percent of time <70 mg/dL as an additional CGM metric. For each GEE model and analysis conducted, covariates such as sex (male/female), BMI (underweight, normal BMI, overweight), exercise type (aerobic, resistance and mixed), exercise duration and age were controlled for. All females were compared with males, all age groups were compared with the youngest age group (6–20 years), the effect of BMI status was assessed via comparison with individuals with normal BMI (18.6–24.9 kg/m^2^), different forms of exercise were compared with aerobic exercise and exercise durations over 20 min were compared with durations under 20 min. All results reported are expressed as interpretations of beta (ß)-coefficients observed in GEE analysis, which refers to the absolute magnitude of difference (delta) in the glucose metric, expressed in mg/dL or as a percent of time, as compared with the reference group after accounting for repeat exercise sessions for any given individual and the other covariates. Data are provided as mean and standard deviation (mean ± SD) or ß-coefficients; [confidence interval]; *p* value, unless otherwise stateds.

## 3. Results

### 3.1. Summary of Descriptive Statistics

The complete database analyzed included a total of 119 individuals (age: 31 ± 21 years, BMI: 22.1 ± 4 kg/m^2^; 77 [65%] females) with a total of 663 exercise events recorded. CGM values during exercise were in the tight glycemic range (i.e., 70–120 mg/dL) for most of the time (85.2 ± 25.7%), with values above this range 9.3 ± 21.6% of the time, and below this range 5.6 ± 16.9% of the time. On average, less than 1% of the CGM data were in the very low range (i.e., 0.6 ± 5.2% <54 mg/dL), while 2.2 ± 9.7% of the values were in the hyperglycemic range (>140 mg/dL). Of the 663 total exercise events, 91 sessions (13.7%) had glucose values <70 mg/dL, with 55 sessions (8.3%) having sustained values of <70 mg/dL for 15 min or more, which is a current criterion for CGM-based diagnosis of hypoglycemia in diabetes [37].

### 3.2. Effects of Exercise and Individual Characteristics on CGM Data Distribution

The influence of sex, age, BMI status, exercise duration and exercise type on CGM data distribution during exercise is shown in Table 1 and Figure 1A–E, with the accompanying GEE analyses provided in Table 2. The subject characteristics for each within-subgroup comparison are shown in Appendix A. Figure 2 displays representative CGM data from study participants within each covariate category (i.e., two representative plots for sex category, four representative plots for age category, three representative plots for BMI category and three representative plots for exercise category).

### 3.3. Effects of Exercise and Individual Characteristics on Acute Glycemic Change Using GEE Analyses

The influence of sex, age, BMI status, exercise type and exercise duration on glucose levels before, during and after exercise is shown in Table 3, while Table 4 highlights these same metrics using GEE analysis.

#### 3.3.1. Sex

Of note, average exercise duration for females was shorter than for males, but only by ~8 min (49 ± 34 min vs. 57 ± 41 min; *p* = 0.009). Mean glucose levels, for all exercise sessions, were 95 ± 17 mg/dL (mean ± SD) vs. 99 ± 15 mg/dL in females and males, respectively (*p* = 0.004 for Student’s t-test), but this sex-related difference was lost with the GEE analyses (Table 4). Nonetheless, females had lower postexercise glycemia than males (93 ± 18 vs. 100 ± 17 mg/dL, ß-coefficient = 4 mg/dL [−9, 0]; *p* = 0.04) and a greater drop in glucose levels during exercise, as measured from pre-exercise to nadir glucose during exercise (14 ± 18 vs. 10 ± 14 mg/dL, ß-coefficient = 5 mg/dL [1, 8]; *p* = 0.009) or from pre- to postexercise (6 ± 20 vs. −1 ± 20 mg/dL, ß-coefficient = 7 mg/dL; [3, 11]; *p* = 0.001). In general, females had a numerically higher % of time <70 mg/dL than males (6.6 ± 18% vs. 3.4 ± 13%; *p* = 0.18) and a lower % of time in the 70–120 mg/dL range (83.7 ± 27% vs. 85.7 ± 26.7%; *p* = 0.09), but these sex-related differences were not statistically significant after accounting for the multiple exercise sessions per subject participant and the other covariates using the GEE analysis. A total of 34 of the 77 women (44%) and 17 of the 42 men (40%) had at least one interstitial glucose value <70 mg/dL during exercise (X^2^ = 0.15, *p* = 0.70), while 20 women (26%) and 9 men (21%) had sustained hypoglycemia (X^2^ = 0.08, *p* = 0.78), as defined by having three or more consecutive CGM values (i.e., 15 min or more) with glucose <70 mg/dL.

#### 3.3.2. Age

Those between the ages of 20 and 39 years had 6.9% more time <70 mg/dL (*p* < 0.001) and 0.6% more time <54 mg/dL (*p* = 0.031) than those aged <20 years. Individuals aged 40–59 years had 8.4% more time in the tight glycemic range (70–120 mg/dL) as compared with those aged <20 years (*p* = 0.012). Individuals aged 40–59 yrs also had significantly less time in the hyperglycemic range, with 6.8% less time >120 mg/dL (*p* = 0.042), 4.3% less time >130 mg/dL (*p* = 0.04) and 2.7% less time >140 mg/dL (*p* = 0.02) than those under the age of 20 years. Using the GEE analysis, it was found that when compared with the youngest age group as reference (age < 20 years), those aged 20–39 years had lower average glucose concentrations during exercise (93 ± 16 vs. 100 ± 16 mg/dL, ß-coefficient = −6 mg/dL [−10, −1]; *p* = 0.03), a lower glucose nadir (83 ± 16 vs. 89 ± 15 mg/dL, ß-coefficient = −6 mg/dL [−10, −1]; *p* = 0.02) and a lower glucose level postexercise (91 ± 21 vs. 100 ± 19 mg/dL, ß-coefficient = −6 mg/dL [−12, 0]; *p* = 0.047). In the individuals aged 40–59 years, average glucose (93 ± 12 mg/dL, ß-coefficient = −5 mg/dL [−10, 0], *p* = 0.04) and peak glucose (102 ± 15 mg/dL, ß-coefficient = −6 mg/dL [−12, 0]; *p* = 0.04) during exercise were also lower than in the youngest age group. Similarly, in the oldest age category (age 60–89 years), average glucose (94 ± 21 mg/dL, ß-coefficient = −7 mg/dL [−14, −1]; *p* = 0.03), peak glucose (104 ± 21 mg/dL, ß-coefficient = −8 mg/dL [−16, 0]; *p* = 0.04) and postexercise glucose (92 ± 21 mg/dL, ß-coefficient = −10 mg/dL [−18 −3]; *p* = 0.009) were all lower, and the rise in glucose from pre to peak glucose during exercise was smaller (5 ± 12 vs. 10 ± 16 mg/dL, ß-coefficient = 6 mg/dL [1, 10]; *p* = 0.008) compared with that of the youngest age group.

#### 3.3.3. BMI

Compared with individuals in the healthy weight BMI category (18.6–24.9 kg/m^2^), those in the underweight category had a higher pre-exercise glucose concentration (104 ± 20 vs. 97 ± 17 mg/dL, ß-coefficient = 6 mg/dL [1, 11]; *p* = 0.02) and a smaller drop in glucose level as measured from baseline to postexercise (3 ± 23 vs. 4 ± 20 mg/dL, ß-coefficient = 5 mg/dL [1, 10]; *p* = 0.03). Individuals in the overweight BMI category group did not display any significant differences in glycemic metrics as compared with those in the healthy weight BMI category group.

#### 3.3.4. Exercise Type

Compared with the aerobic exercise sessions, pre-exercise glucose was higher in the resistance exercise sessions (99 ± 18 vs. 93 ± 13 mg/dL, ß-coefficient = −4 mg/dL [−8, 0]; *p* = 0.02), while the drop in glucose from pre-exercise to nadir glucose was smaller (7 ± 12 mg/dL vs. 12 ± 18 mg/dL, ß-coefficient = −6 mg/dL [−10, −2]; *p* = 0.005). In the mixed exercise sessions, greater reductions in glycemia were observed relative to the aerobic exercise sessions, as measured from pre to nadir (20 ± 17 vs. 12 ± 18 mg/dL, ß-coefficient = 6 mg/dL [11, 5]; *p* = 0.029) or from pre- to postexercise (15 ± 20 vs. 3 ± 21 mg/dL, ß-coefficient = 11 mg/dL [5, 16]; *p* < 0.001), while the rise in glucose from pre-exercise to peak glucose concentration was smaller in the mixed sessions as compared with that in the aerobic exercise sessions (2 ± 12 vs. 8 ± 15 mg/dL, ß-coefficient = 6 mg/dL [1, 10]; *p* = 0.008). Resistance exercise was associated with 0.7% less time <54 mg/dL (*p* = 0.04), 2.7% less time >130 (*p* = 0.03) and 1.6% less time >140 mg/dL (*p* = 0.027) as compared with the aerobic exercise sessions.

#### 3.3.5. Exercise Duration

Although pre-exercise glucose level and average glucose levels were similar among the various exercise duration groupings, averaging 98.5 ± 18.4 mg/dL and 96.2 ± 16.1 mg/dL, respectively, for all sessions combined, the increasing exercise duration categories (i.e., 21–39; 40–59; 60–89; 90–120 and 120–300 min) had lower nadirs (all *p* < 0.001), higher peaks (*p* = 0.025; *p* = 0.026; *p* = 0.001; *p* < 0.001) and greater changes in glycemia during exercise as measured from pre to nadir (all *p* < 0.001), pre to peak (*p* = 0.222; *p* = 0.185; *p* = 0.004; *p* = 0.004; *p* < 0.001) or peak to nadir (all *p* < 0.001), as compared with the shortest exercise duration category (10–20 min).

## 4. Discussion

As CGM accessibility, accuracy and wearability improve [39], the use of this technology is steadily increasing in the apparently healthy (i.e., nondiabetic) [21,22,23,24,25,26,27,40]. CGM use has been touted by new and emerging companies to help apparently healthy individuals learn more about how their glycemia responds to lifestyle choices including exercise and dietary habits. However, the use of CGM by people not living with diabetes has been questioned by some [18], perhaps because it is generally felt that circulating blood glucose is managed in a relatively tight range (i.e., 70–120 mg/dL) most of the time, even during most forms of prolonged exercise [1]. Indeed, various companies and wellness programs that use CGM for personal weight loss and nutritional counselling for persons without diabetes (e.g., NutriSense, Levels [41,42]) suggest that, ideally, fasting CGM values should range between ~70 and 85 mg/dL, with a post-meal glucose <110 mg/dL. Another emerging company that focuses on exercise training and endurance competitions (Supersapiens [43]) suggests that one’s individualized glucose “performance” zone may be higher for some exercise events to optimize fuel provision (~110–130 mg/dL) (https://www.supersapiens.com [accessed on 24 October 2023]). However, the normal distribution of glucose values within these various glucose zones during meals and exercise is unclear.

Shah and colleagues [25] found that nondiabetic individuals had ~90% time between 70 and 120 mg/dL over an average ten-day CGM use period, with a median time <70 mg/dL of only 1.1%, while DuBose et al. found that mean glucose levels rose from 93 ± 10 mg/dL to 130 ± 13 mg/dL with meals and dropped by 15 ± 18 mg/dL and 9 ± 12 mg/dL with aerobic and resistance exercise, respectively [26]. This current study reports more detailed exercise CGM data from this same cohort of healthy individuals not living with diabetes. Our analysis was limited to exercise events lasting 10 min or more, because of technical limitations of the CGM data-reporting format (values for this sensor report every 5 min), that followed 30 min or more after a meal to help minimize the potential for meal-related confounders on CGM metrics. While we show that healthy individuals spent most of the time (84 ± 27%) with their glucose in the so-called “tight” glycemic range of 70–120 mg/dL during exercise, we also report here for the first time that several participant-level (i.e., sex, age, BMI status) and event-level (i.e., exercise type, exercise duration) factors influence CGM values during exercise in apparently healthy individuals. Stratifying these data based on these variables may be useful as a benchmark for the development of new technologies and strategies that aim to optimize glucose control during exercise for men and women or for those who are in a certain age and/or BMI category. With newer CGM systems that report glucose metrics at 1 min intervals, shorter exercise activities such as exercise snacks [44] could also be investigated.

Surprisingly, the impact of sex on interstitial glucose levels during exercise is an under-investigated topic, even in the diabetes literature [45]. In healthy individuals not living with diabetes, women have been reported to have higher blood (i.e., venous) glucose levels than men during aerobic exercise in some [4,13] but not all [14,15,16] laboratory-based studies. When CGM was used in an earlier study of 34 nondiabetic adult participants, women tended to have a greater percent of time with glucose <70 mg/dL than men (women 3.2 ± 3.0% vs. men 1.7 ± 1.5%, *p* < 0.01), but that study was not focused on exercise and used a first-generation CGM device that may have overreported glucose values < 70 mg/dL [33]. In this study of 663 exercise sessions performed by 119 apparently healthy men and women with no diagnosed dysglycemia as assessed based on a point-of-care HbA1c level <5.7% (39 mmol/mol), we found that women tend to have a higher percent of time <70 mg/dL as compared with men (6.6 ± 18 vs. 3.4 ± 13%), as well as modestly lower postexercise glycemia than men (by 4 mg/dL) and greater reductions in glucose during exercise, as measured either via pre- to postexercise changes in glucose or via pre to nadir glucose. However, after accounting for multiple covariates including the number of repeat exercise events for a given individual, BMI, age and exercise type and duration, we failed to observe any clear sex-specific differences in any of the percents of time within various CGM target zones, except for percent of time within target, which tended to be lower in females than males, by about 5% (*p* = 0.05) (Table 2). These findings suggest that females may have a slightly lower percent of time in the tight glucose target range (70–120 mg/dL) as compared with males, with potentially more exposure to exercise-induced hypoglycemia (as measured using % of time below range) as compared with males, even when their exercise duration may be shorter. Our findings conflict somewhat with the repeated observations that women tend to rely more on lipids as fuel than carbohydrates during exercise [46], but are possible given that women tend to have lower glucagon levels and reduced hepatic glucose production during exercise as compared with men [47]. Given the inconsistencies in the findings, and the likely small effect size between the sexes, further research with larger, more diverse datasets is needed to clarify if women may be more at risk for exercise-associated hypoglycemia then men, particularly in longer endurance events.

Age markedly influences percent of time in the universally accepted clinical target range for people living with diabetes (i.e., 70–180 mg/dL). More specifically, younger individuals with diabetes tend to have more hyperglycemia and a lower percent of time in either “tight” (70–140 mg/dL) or more “relaxed” (70–180 mg/dL) clinical target ranges for diabetes, as compared with older individuals living with the disease [48]. However, the impact of age on CGM in healthy individuals not living with diabetes is less clear. Shah and colleagues [25] found that individuals ≥60 years of age had nearly double the percent of time above 140 mg/dL when compared with those aged <60 years. Interestingly, we found that the youngest (<20 years) and the oldest (age > 60 years) individuals tended to have the highest percents of time above the tight glycemic range as compared with the young adult cohort (ages 20–39 years), with the youngest group having elevated pre-, during- and postexercise glycemia. The reasons for higher glucose levels during exercise in the youngest age group are unclear but could be related to the nature of their activity, which tends to be more intermittent and vigorous in nature as compared with that of adults [49], or because of the hormonal changes associated with growth and development and a higher rate of lipid oxidation during exercise [50]. The middle-aged individuals (age 40–59 years) in this cohort had the highest percentage of time in the range, with the lowest amounts of time below and above the range. Thus, overall, younger individuals (age < 20 years) appear to have a tendency for higher glucose levels during exercise, with less hypoglycemia risk, as compared with the other age groups, but there is no obvious linear effect of age on any of the CGM category metrics. In fact, the longest time in the tight range, with the lowest hypoglycemia and hyperglycemia risks, appears to occur in the middle age group (age 40–59 years). The reasons why younger individuals have a higher percent of time above range than two of the older age groups are unclear and require confirmation with a larger dataset that includes nutritional data and information about competition status.

In our analyses, BMI status was found to have little impact on glycemia during exercise, except that pre-exercise glycemia was highest in the underweight BMI group, which likely helps to explain the greater changes in glucose pre- to postexercise in that category as compared with the healthy weight BMI group. Based on HbA1c levels and/or frequent glucose measurements in plasma, underweight women also display more glucose instability [51] and a greater risk for elevations in glycemia and prediabetes in spite of having a lean body mass [52]. In our study, the highest BMI category (i.e., overweight) had similar glycemia prior to and during exercise as compared with the healthy weight BMI category group, suggesting that being overweight does not necessarily influence exercise-related glycemia all that much. This is expected, since otherwise healthy overweight individuals typically do not present with dysglycemia in the postabsorptive state, at least when glucose is measured at rest after a test meal [53]. Moreover, overweight women, who were deemed at elevated risk for type 2 diabetes, were essentially identical to normal-weight women with respect to CGM analysis, both prior to and during high-intensity interval-type exercise and during moderate-intensity aerobic exercise [54].

To date, few studies have determined if exercise type (i.e., aerobic, resistance, mixed) differentially influences CGM-based glycemia in healthy individuals. We found that resistance exercise was associated with the highest percent of time in the tight glycemic range (i.e., 70–120 mg/dL), the lowest percent of time <54 mg/dL and the smallest reduction in glycemia as measured from pre to nadir or peak to nadir glucose concentration. Resistance exercise also appears to result in lower percents of time >130 mg/dL and >140 mg/dL as compared with aerobic exercise sessions. We also found that mixed forms of exercise were associated with greater changes in glycemia, by as much as 11 mg/dL, as measured using pre to post, pre to nadir or pre to peak glycemia as compared with aerobic-only exercise. Although CGM values <70 mg/dL were observed with all forms of exercise, the highest percent of time ≤70 mg/dL was noted with mixed exercise (12.2 ± 26.6%). As expected, we found lower glucose nadirs as the exercise duration increased, but we also observed higher glucose peaks. This may be because with a greater exercise duration, there is more opportunity to observe more extreme glucose events when using CGM, some of which may be sensor errors. However, it is also known that ultralong exercise durations (>2 h) can be associated with clinical hypoglycemia if carbohydrate feeding is limited [7,8,9] or mistimed [5,6], and elevations in glycemia can also be observed if excessive carbohydrate loading occurs [10,11]. As such, CGM may be a particularly useful tool to notify active individuals when they should (or should not) be fueling with simple carbohydrates, particularly during more prolonged activities.

This analysis has a number of strengths and limitations that should be acknowledged. A major strength of this study is that it contains a relatively large dataset of CGM-based glucose values reported during exercise for healthy individuals without diabetes. Moreover, this is a free-living study that allows observations to be made on exercise glycemia in a less restrictive environment than previous laboratory-based studies, which typically observed individuals in a fasted state or with standardized carbohydrate feeding in a highly controlled laboratory environment. In terms of limitations, the analyses included in this study did not take into account a specific time of day for exercise sessions, the types of foods and/or beverages consumed during exercise (if any) or the relative exercise intensity of the activity, which all likely impact the glucose response to activity [55,56]. Moreover, only exercise activities lasting 10 min or more were included. Additionally, specific insertion sites for CGM devices have not been recorded, although sensor placement may impact exercise accuracy to some degree, even in individuals without diabetes [22]. For some of the comparisons made, the number of subject participants and exercise sessions within each subcategory may have been insufficient to detect true differences or the subgroupings may have been “unbalanced” with respect to other potential confounders (Appendix A). It is worth noting that there were baseline differences in the group of individuals performing predominantly resistance exercise, which may impact our observations. Moreover, we were unable to determine if glycemia impacted exercise performance, which should be a focus of future research. Lastly, it should be emphasized that we performed multiple comparisons in our exploratory analysis of the secondary outcome measures without adjusting for the potential for false discoveries. However, these comparisons were prespecified, and the GEE model is more robust than more traditional analysis of variance because of its ability to exert an analytical focus specifically on group differences while also accounting for clustering, which helped to account for the uneven repeat exercise sessions between participants [57]. Future studies should include an assessment of the potential influence of meals (i.e., macronutrient status and timing), exercise training status and relative exercise intensity during exercise, which all likely influence glycemia during exercise. The accuracy of CGM devices during exercise, including exercise in different ambient settings, and with different sensor placements, should also be investigated.

In summary, ~85% of all CGM values are within a relatively narrow glycemic range (70–120 mg/dL) during exercise in healthy individuals, with the remaining CGM data split between low glucose (~5.5% <70 mg/dL) and high glucose (%9.5, >120 mg/dL) readings. Overall, in a healthy (but not athletic) cohort of physically active individuals, women tend to have a higher percent of time <70 mg/dL and greater reductions in glycemia during exercise as compared with men and younger individuals (age <20 years), and those with an underweight BMI status and those of a young age (i.e., <20 years) may have higher pre-exercise glucose concentration, higher average glucose levels during exercise and higher immediate postexercise glycemia, albeit the effect sizes may be small. In general, in apparently healthy individuals, resistance exercise appears to have less risk for low glucose levels and less exposure to high glucose levels as compared with aerobic exercise, while mixed forms of exercise tend to have the greatest likelihood of a drop in glycemia and a higher percent of time with glucose ≤70 mg/dL. Future studies should focus on the effects of relative exercise intensity, using wearable technologies, nutrition, exercise time of day and other factors on CGM metrics during exercise in the nondiabetic population, with the possible goal of improving metabolic health and/or performance in the general population.

## Figures and Tables

**Figure 1 sensors-23-09059-f001:**
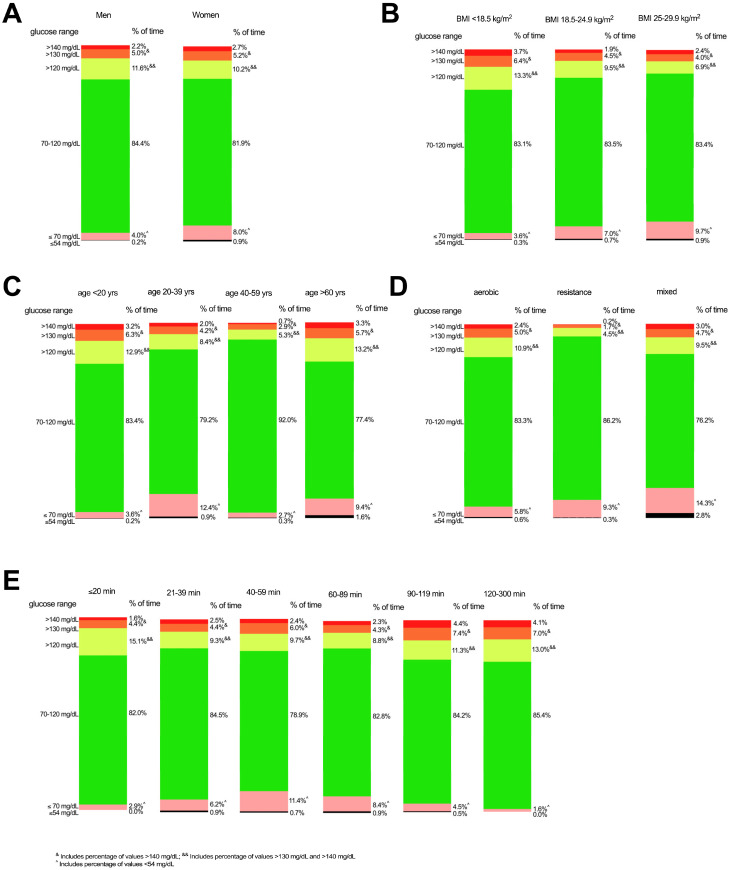
Percent of time in range, time below range and time above range for CGM values based on sex (**A**), age (**B**), BMI (**C**), exercise type (**D**) and exercise duration (**E**) during exercise. Black bars represent percent of time <54 mg/dL; pink bars represent percent of time <70 mg/dL; green bars represent percent of time 70–120 mg/dL; light green bars represent percent of time >120 mg/dL; orange bars represent percent of time >130 mg/dL; red bars represent percent of time >140 mg/dL. Means and standard deviations are shown in Table 1. Note: Percentages shown for values <70 mg/dL also include the values <54 mg/dL. Similarly, percentages shown for values >120 also include the values for >130 mg/dL and >140 mg/dL, while percentages >130 mg/dL also include the values for >140 mg/dL. ^ *p* < 0.05; ^&&^ *p* < 0.01 based on GEE analysis (see Table 2). For participant characteristics within each subcategory, see Appendix A.

**Figure 2 sensors-23-09059-f002:**
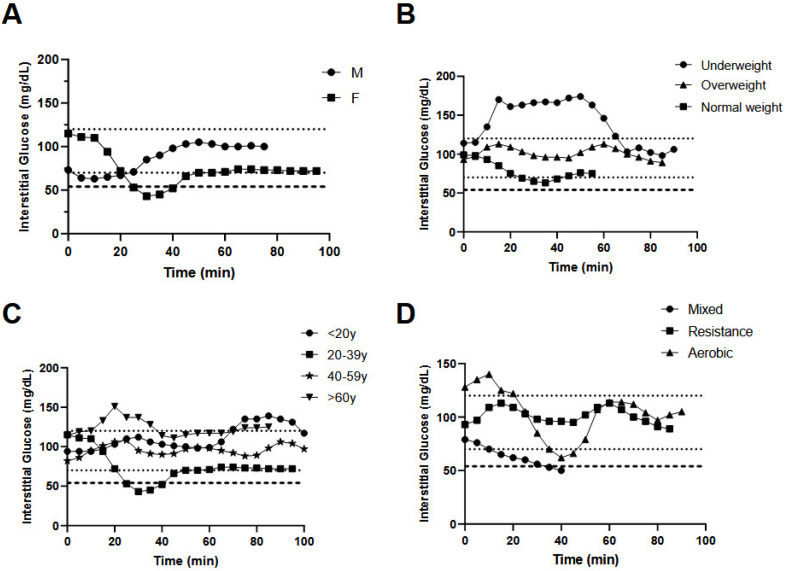
Representative plots of glycemia over time during exercise sessions categorized by sex (**A**), BMI (**B**), age (**C**) and exercise type (**D**). Line graphs in panels (**A**–**D**) are plots of changes in interstitial glucose concentrations over time throughout the exercise duration. These depicted changes are of one participant representing each group under every category (sex, BMI, age, exercise type) used as a covariate.

**Table 1 sensors-23-09059-t001:** **Mean time in range values (% time) *in various CGM target ranges*.** Values are means ± SD and expressed in percentage of time in each range (% time) for all exercise sessions within each category. Data are represented in graphical format in Figure 1. Greyed-out rows represent the categories used as reference ranges for subsequent GEE model multivariate analysis (see Table 2 for GEE analyses on time in range).

Covariate	Categories	<54 mg/dL	<70 mg/dL	70–120 mg/dL	>120 mg/dL	>130 mg/dL	>140 mg/dL
Sex	Males (n = 228)	0.1 ± 1.7	3.4 ± 13.4	85.7 ± 26.6	10.8 ± 24.3	4.7 ± 15.2	2.0 ± 9.1
	Females (n = 435)	0.8 ± 7.0	6.6 ± 18.0	83.8 ± 26.8	9.6 ± 22.6	5.1 ± 16.1	2.7 ± 11.2
Age (years)	6–19 (n = 276)	0.2 ± 1.9	2.9 ± 11.0	85.0 ± 25.2	12.1 ± 23.8	5.9 ± 16.8	2.9 ± 11.3
	20–39 (n = 156)	0.8 ± 5.3	10.3 ± 22.8	82.2 ± 28.3	7.6 ± 20.5	4.1 ± 13.1	2.1 ± 8.8
	40–59 (n = 110)	0.2 ± 1.7	2.5 ± 10.8	91.9 ± 20.0	5.6 ± 17.6	2.6 ± 10.7	0.5 ± 3.6
	60–89 (n = 121)	1.4 ± 9.4	8.1 ± 20.3	79.5 ± 31.8	12.4 ± 28.2	6.2 ± 19.8	3.6 ± 13.9
BMI (kg/m^2^)	<18.6 (n = 173)	0.3 ± 2.3	3.1 ± 12.0	84.0 ± 26.3	13.0 ± 24.2	6.2 ± 17.4	3.5 ± 12.8
	18.6–24.9 (n = 327)	0.6 ± 5.9	5.8 ± 16.7	85.4 ± 25.7	8.8 ± 21.9	4.3 ± 14.2	1.8 ± 8.5
	25–29.9 (n = 139)	0.8 ± 5.4	8.0 ± 21.1	85.3 ± 26.8	6.7 ± 19.5	3.7 ± 13.6	2.4 ± 10.7
Exercise type	Aerobic (n = 534)	0.5 ± 4.0	4.9 ± 15.3	84.9 ± 25.8	10.2 ± 22.8	4.9 ± 15.3	2.3 ± 10.0
	Resistance (n = 51)	0.3 ± 2.0	7.7 ± 21.4	88.2 ± 24.9	4.1 ± 15.1	1.2 ± 5.3	0.2 ± 1.6
	Mixed (n = 39)	2.7 ± 13.9	12.2 ± 26.6	77.9 ± 34.0	9.8 ± 25.2	4.3 ± 14.0	2.9 ± 10.4
Exercise duration (minutes)	10–20 (n = 105)	0.0 ± 0.0	2.8 ± 11.5	84.3 ± 30.3	12.9 ± 29.3	5.1 ± 17.6	1.4 ± 8.7
	21–39 (n = 179)	0.8 ± 6.2	5.1 ± 17.5	85.9 ± 27.4	9.0 ± 22.8	4.5 ± 16.8	2.6 ± 12.3
	40–59 (n = 117)	0.5 ± 3.0	9.4 ± 20.1	81.0 ± 27.4	9.6 ± 22.3	5.3 ± 16.1	2.3 ± 10.2
	60–89 (n = 161)	0.9 ± 7.1	7.2 ± 19.2	84.4 ± 26.6	8.4 ± 21.2	4.2 ± 13.6	2.3 ± 9.3
	90–120 (n = 46)	0.4 ± 2.9	3.2 ± 8.5	85.6 ± 19.5	11.1 ± 19.2	6.5 ± 14.3	2.2 ± 8.0
	120–300 (n = 55)	0.0 ± 0.0	1.2 ± 4.4	86.3 ± 21.7	12.5 ± 21.7	6.7 ± 15.9	4.1 ± 12.9

**Table 2 sensors-23-09059-t002:** GEE multivariable analysis for the percent of time in various CGM target ranges relative to the reference groups as shown in Table 1. Values are shown as the ß-coefficient (% time); *p* value relative to each of their respective reference groups (i.e., male; normal BMI; aerobic exercise only; exercise duration ≤20 min; age ≤19 years). Values shown in bold highlight statistical significance.

Covariate	Categories	≤54 mg/dL	≤70 mg/dL	71–119 mg/dL	≥120 mg/dL	≥130 mg/dL	≥140 mg/dL
Sex	Females	0.3, 0.26	2, 0.15	−5, 0.06	2, 0.34	2, 0.22	1, 0.25
Age (years)	20–39	**0.6,** **0.04**	**8,** **0.002**	−3, 0.51	−5, 0.18	−3, 0.19	−1, 0.37
	40–59	−0.2, 0.60	−3, 0.29	**11,** **0.003**	**−8,** **0.02**	−5, 0.05	**−3,** **0.02**
	60–89	2, 0.29	6, 0.16	2, 0.66	**−8,** **0.03**	**−6,** **0.03**	−2, 0.38
BMI	Underweight	0.3, 0.17	1, 0.57	−1, 0.79	−0.1, 0.97	−1, 0.75	1, 0.64
	Overweight	0.4, 0.60	3, 0.41	−3, 0.35	0.5, 0.87	1, 0.51	2, 0.22
Exercise type	Resistance	**−0.8,** **0.04**	0.4, 0.92	4, 0.41	−4, 0.17	−2, 0.12	**−2,** **0.04**
	Mixed	2, 0.32	7, 0.20	−8, 0.12	1, 0.78	0.2, 0.93	1, 0.58
Exercise duration (minutes)	21–39	0.8, 0.11	3, 0.17	−1, 0.72	−1, 0.69	2, 0.33	1, 0.23
	40–59	0.3, 0.56	5, 0.06	−6, 0.21	1, 0.89	**4,** **0.05**	1, 0.29
	60–89	0.6, 0.05	4, 0.13	−3, 0.44	−1, 0.88	3, 0.14	2, 0.13
	90–120	0.1, 0.89	0.2, 0.96	−0.4, 0.93	0.2, 0.96	5, 0.05	**3,** **0.05**
	120–300	0.1, 0.44	−1, 0.65	−1, 0.76	2, 0.63	5, 0.10	3, 0.13

**Table 3 sensors-23-09059-t003:** Mean CGM values (mg/dL) before (pre), during (average, nadir, peak) and after (post) exercise, along with change (delta) in glucose level from pre to nadir, pre to post, pre to peak and peak to nadir according to covariate status. Values are means ± SD and expressed in mg/dL for all exercise sessions within each category. Greyed-out rows represent the categories used as reference ranges for subsequent GEE model multivariate analysis (see Table 4 for GEE analyses). Numbers in brackets for each category indicate (# of participants/# of sessions) for males and females, or (# of sessions) for all other categories.

Covariate	Categories	Pre	Average	Nadir	Peak	Post	ΔPre–Nadir	ΔPre–Peak	ΔPeak–Nadir	ΔPre–Post
Sex	Males (42/227)	99 ± 18	99 ± 15	89 ± 15	109 ± 20	100 ± 20	10 ± 16	−10 ± 17	20 ± 19	−1 ± 20
	Females (77/436)	98 ± 19	95 ± 17	85 ± 16	105 ± 21	92 ± 19	14 ± 18	−6 ± 14	20 ± 17	6 ± 20
Age (years)	1–19 (276)	101 ± 19	100 ± 16	89 ± 15	111 ± 21	100 ± 19	12 ± 19	−10 ± 16	22 ± 19	1 ± 21
	20–39 (155)	96 ± 17	93 ± 16	83 ± 16	103 ± 21	91 ± 21	14 ± 18	−6 ± 16	20 ± 19	6 ± 22
	40–59 (109)	95 ± 14	93 ± 12	85 ± 12	102 ± 15	92 ± 14	10 ± 12	−7 ± 12	17 ± 12	3 ± 16
	60–89 (120)	99 ± 21	94 ± 19	85 ± 20	104 ± 21	92 ± 21	14 ± 16	−5 ± 12	19 ± 15	8 ± 19
BMI	Underweight (173)	104 ± 20	101 ± 16	90 ± 15	113 ± 22	100 ± 19	14 ± 21	−9 ± 17	23 ± 21	3 ± 23
	Normal weight (349)	97 ± 17	95 ± 16	85 ± 15	105 ± 19	94 ± 19	12 ± 16	−8 ± 14	20 ± 15	4 ± 20
	Overweight (138)	96 ± 17	93 ± 15	84 ± 15	102 ± 20	90 ± 19	12 ± 15	−5 ± 14	18 ± 17	6 ± 20
Exercise type	Aerobic (534)	99 ± 18	97 ± 16	87 ± 16	107 ± 21	96 ± 19	12 ± 18	-8 ± 15	20 ± 18	3 ± 21
	Resistance (51)	93 ± 13	93 ± 13	85 ± 14	101 ± 16	92 ± 18	7 ± 12	−8 ± 14	15 ± 13	.5 ± 16
	Mixed (38)	100 ± 17	93 ± 18	81 ± 18	105 ± 21	87 ± 22	20 ± 17	−2 ± 12	24 ± 17	15 ± 20
Exercise duration (minutes)	10–20 (105)	99 ± 17	100 ± 16	97 ± 16	103 ± 17	100 ± 17	3 ± 9	−4 ± 10	7 ± 7	−1 ± 12
	21–39 (178)	97 ± 19	95 ± 16	87 ± 16	102 ± 19	93 ± 18	9 ± 16	−5 ± 13	15 ± 13	4 ± 21
	40–59 (116)	100 ± 19	94 ± 16	83 ± 16	106 ± 20	91 ± 19	17 ± 18	−6 ± 12	23 ± 16	10 ± 21
	60–89 (160)	97 ± 19	94 ± 16	82 ± 15	106 ± 22	94 ± 21	16 ± 18	−8 ± 16	24 ± 17	5 ± 21
	90–120 (45)	99 ± 14	98 ± 15	84 ± 14	115 ± 23	97 ± 22	15 ± 18	−15 ± 22	31 ± 21	1 ± 24
	120–300 (54)	102 ± 20	102 ± 14	85 ± 11	121 ± 21	102 ± 17	17 ± 20	−21 ± 20	36 ± 21	−0.02 ± 25

**Table 4 sensors-23-09059-t004:** GEE multivariable analysis for the CGM-derived metrics relative to the reference groups as shown in Table 3. Values are shown as the ß-coefficient (mg/dL) rounded to the nearest whole integer; *p* values are relative to each of their respective reference groups (i.e., male; normal BMI; aerobic exercise only; exercise duration ≤20 min; age ≤19 years). Values shown in bold highlight statistical significance.

Covariate	Categories	Pre	Average	Nadir	Peak	Post	Δ Pre–Nadir	Δ Pre–Peak	Δ Peak–Nadir	Δ Pre–Post
Sex	Females	2; 0.45	−1; 0.74	−3; 0.10	0; 0.86	**−4**; **0.04**	**5**; **0.009**	1, 0.43	3, 0.06	**7**; **0.001**
Age (years)	20–39	−3; 0.19	**−6**; **0.03**	**−6**; **0.02**	−6; 0.08	**−6**; **0.05**	3; 0.26	3, 0.09	0, 0.99	4; 0.14
	40–59	−4; 0.09	**−5**; **0.04**	−3; 0.22	**−6**; **0.04**	−5; 0.10	−2; 0.43	2, 0.46	−3, 0.12	0.3; 0.94
	60–89	−2; 0.52	**−7**; **0.03**	−5; 0.09	**−8**; **0.04**	**−10**; **0.009**	2; 0.47	**6**, **0.02**	−3, 0.27	7; 0.06
BMI	Underweight	**6**; **0.02**	2; 0.35	0.6; 0.77	4; 0.21	−0.04; 0.99	5; 0.05	2, 0.32	3, 0.17	**5**; **0.03**
	Overweight	2; 0.47	0; 0.91	0; 0.89	1; 0.83	−2; 0.56	1; 0.58	1, 0.68	0, 0.85	3; 0.29
Exercise type	Resistance	**−4**; **0.04**	−1; 0.57	1; 0.64	−4; 0.22	−1; 0.70	**−6**; **0.005**	−1, 0.68	**−5**, **0.008**	−3; 0.28
	Mixed	3; 0.36	−1; 0.86	−2; 0.50	0; 0.95	−6; 0.10	**6**; **0.03**	**6**, **0.008**	2, 0.45	**11**; <**0.001**
Exercise duration (minutes)	21–39	−2; 0.42	−4; 0.09	**−8**; **<0.001**	0; 0.95	**−6**; **0.02**	**6**; **<0.001**	−2, 0.22	**8**, **<0.001**	3; 0.07
	40–59	5; 0.09	−2; 0.36	**−11**; <**0.001**	**6**; **0.03**	**−6**; **0.02**	**15**; **<0.001**	−2, 0.19	**17**, **<0.001**	**11**; **<0.001**
	60–89	0; 0.95	−3; 0.20	**−12**; **<0.001**	**6**; **0.03**	−3; 0.32	**13**; **<0.001**	**−5**, **0.004**	**18**, **<0.001**	4; 0.13
	90–120	1; 0.61	0; 0.98	**−12**; **<0.001**	**12**; **0.001**	−1; 0.88	**13**; **<0.001**	**−10**, **0.004**	**24**, **<0.001**	2; 0.66
	120–300	5; 0.23	3; 0.29	**−10**; **<0.001**	**18**; **<0.001**	1; 0.68	**16**; **<0.001**	**−16**, **<0.001**	**29**, **<0.001**	3; 0.49

## Data Availability

The data that support the findings of this study are available from the corresponding author upon reasonable request.

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
