# Peer review of "The Impact of Sex, Body Mass Index, Age, Exercise Type and Exercise Duration on Interstitial Glucose Levels during Exercise"

_sensors, 2023, doi:10.3390/s23229059_

Round 1

Reviewer 1 Report

Comments and Suggestions for Authors

see the attach file.

Author Response

We would like to take this opportunity to thank the reviewers for their insightful comments on our original submission. Your comments have been particularly valuable in helping to improve our paper. We have itemized our responses and revisions below.

Reviewer comments

REVIEWER 1:

The title is acceptable.

Abstractsummarizes the main content of the paper, but the abstract is too generalized. Conclusion in abstract must be rewritten.

Thank you for this comment. Unfortunately, due to the 200-word limit, we were unable to add more detail to the one paragraph abstract. We have instead added more detail to the main conclusion as recommended in the comments below.

Introduction- The general overview of the interstitial glucose levels during exercise is summarized. It would be better if more succinct and clear, and add the necessity of writing this article.

In addition to applications listed in the introduction, we have further clarified the purpose/application of this article at the end of the introduction:

We feel that this large dataset will aid in establishing set normative ranges for individuals without diabetes while factoring in covariates of age, sex, exercise type and duration. It is our intent to establish these normative ranges for further application in future studies involving glycemic management with and without diabetes.

Methods- The author obtains some stale knowledge and conclusions by reading a lot of old literature. Such studies should inform readers of the latest, most authoritative research rather than confuse them with old conclusions. Furthermore, no information is added through which software GEE analysis were conducted.

We have looked at our references and done a recent search on this topic. We have added several new references, highlighted in red in our revised reference list. We welcome any other papers we may have missed. We now also note that the software used for GEE analysis has been listed (IBM SPSS Statistical Software).

Results- Well written but no proper interpretations of GEE analysis are provided. Even in tables no significance level is mentioned. Only mean and standard deviation is mentioned which is not enough

Thank you for this comment. We have expanded our explanation of the GEE method, which is limited to tables 2-4. Table 1 is just descriptive. Tables 2 and 4 outline statistical significance of βco-efficient values with significant values highlighted in bold. Tables 3 and 5 were indicative of the mean values. We have now moved the means tables to supplementary data to prevent confusing the reader.

References- The number of references is sufficient, but most of the references are very old, which fails to provide readers with the latest research results and ideas in this field

We appreciate this feedback...We have updated the references after conducting another search on the topic looking for more related papers

Conclusion- The conclusion is written confusingly. Must be rewritten with more clarity.

The conclusion in the manuscript is revised. We have also added a few edits and some future directions to the conclusion of the manuscript.

Tables and figures- The tables and figures are clear and exquisite

General Comments and Decision-Must be revised.

Our revision has been drafted and uploaded. Thank you for the opportunity to submit an improved manuscript based on your excellent feedback.

Reviewer 2 Report

Comments and Suggestions for Authors

The manuscript presents an interesting subject of several agents impact on interstitial glucose levels during exercise. Indeed, measurement of sex, age, BMI on interstitial glucose levels by continuous glucose monitoring (CGM) during exercise (type and duration) in the general population seems unexplored.

The language is correct throughout the paper and the references are up to date.

However, a spontaneous limitation during reading this article is that a relatively large dataset of CGM-based glucose values reported during exercise for healthy individuals without diabetes, in a free-living study of a less restrictive environment that previous studies, where analyses did not take into account time of day for exercise sessions, any foods consumed during exercise, which all impact the glucose response to activity. Moreover, it was no possibility to determine if glycemia impacted exercise performance. If the authors could add 2-3 lines-sentences in the discussion, in the same context, of their opinion about an assessment of the potential influence of meals, macronutrient status and timing, exercise training status and relative exercise intensity during exercise which all influence glycemia during exercise, in order to lead and prevent the findings of several future studies. The authors are also encouraged to add 2-3 sentences of their personal opinion, based on their expertise, on the accuracy of CGM devices during exercise, including exercise in different ambient settings, and with different sensor placements, in order to attempt making an intermediate smoother step transition between this work to future studies in this direction.

Author Response

REVIEWER 2:

The language is correct throughout the paper and the references are up to date.

However, a spontaneous limitation during reading this article is that a relatively large dataset of CGM-based glucose values reported during exercise for healthy individuals without diabetes, in a free-living study of a less restrictive environment that previous studies, where analyses did not take into account time of day for exercise sessions, any foods consumed during exercise, which all impact the glucose response to activity. Moreover, it was no possibility to determine if glycemia impacted exercise performance. If the authors could add 2-3 lines-sentences in the discussion, in the same context, of their opinion about an assessment of the potential influence of meals, macronutrient status and timing, exercise training status and relative exercise intensity during exercise which all influence glycemia during exercise, in order to lead and prevent the findings of several future studies.

We have included the following clarification in the methods section of the article:

Our comparisons (age, BMI, sex, exercise type, exercise duration) were pre-defined. Time of day for all sessions was between 5am-12am midnight but no sub-analyses was performed. Exercise sessions were only included for sessions that involved meals limited to breakfast, lunch and dinner with no other meals recorded during the day. This is in keeping with the inclusion criteria also supported in the previously published study by Dubose et al 2020 which was the primary study that collected the data we have now analyzed in this research article. We agree that these additional research questions should be examined, perhaps in larger and newer datasets. We believe Supersapiens may have a more appropriate data base for this given the larger database of mostly European users.

The authors are also encouraged to add 2-3 sentences of their personal opinion, based on their expertise, on the accuracy of CGM devices during exercise, including exercise in different ambient settings, and with different sensor placements, in order to attempt making an intermediate smoother step transition between this work to future studies in this direction.

Rather than give our own personal impressions (we think they are reasonably accurate but comparing against blood glucose is a real problem because of time delays in glucose equilibrium), we have highlighted a few good studies that show that CGM accuracy is not as good during exercise as during rest. Accuracy does appear to be a little better when the CGM is placed on the arm vs elsewhere on the body in people not living with diabetes, according to a one recent papers Burr study (Coates AM, Cohen JN, Burr JF. Investigating sensor location on the effectiveness of continuous glucose monitoring during exercise in a non-diabetic population. Eur J Sport Sci. 2023;23(10):2109–17.).

Reviewer 3 Report

Comments and Suggestions for Authors

Dear Editor,

In regard to the manuscript ID- sensors-2617180 entitled The impact of sex, body mass index, age, exercise type and exercise duration on interstitial glucose levels during exercise Short running title: CGM during exercise in healthy adults”.  The authors studied the impact of age, sex, body mass index on interstitial glucose levels as measured by continuous glucose monitoring (CGM) during exercise in the healthy population concluding that the studied factors influence the CGM measurements. The authors explain several points about the considerations during their study, however, there are some experimental conditions, results, and discussion that need to be clarified, before considering the manuscript for its publication at sensors.

Could the authors include information about the exercise that was performed and the duration? The authors mentioned that the duration was more than 10 minutes but, could the authors explain why?

The authors showed the results in Tables, could the authors explain and organize them in a more understandable format and use the same significant figures, in the same regard, could the authors explain better the figure captions?

The authors showed the results in tables, could the authors include representative plots of the glucose change in time for at least one participant in each study group?

In the main text, the authors need to verify the format of each numerical result, seems that for several results something is missing between the 2 numerical data.

The authors need to verify the format of their references in the main text, some of the references are in parenthesis, and some others square brackets.

Could the authors include information about the influence of the temperature during their studies?

The authors need to organize in a better form their results.

The reviewer recommends major revisions before considering the manuscript for publication at Sensors.

Sincerely,

The reviewer.

Author Response

Could the authors include information about the exercise that was performed and the duration? The authors mentioned that the duration was more than 10 minutes but, could the authors explain why?

This is a valid point and limitation of our work that we did not explain very well..The CGM unit used in this study only reports glucose values every 5 minutes (in fact the unit measure more frequently than every 5 minutes, apparently, but then the device only shows a 5-min average value after the internal Dexcom proprietary algorithm filters the data), so we felt two values would be needed to make any conclusions, so a min bout of 10 minutes was pre-selected for this study. Future work can now be done with newer sensors that report data every 60 seconds, but that was unavailable at the time of data collection.

We used the study application to classify all activities as aerobic, resistance or mixed format, given this is typically how CGM data is clustered for people living with diabetes (this has been referenced in the revised methods section). An exercise duration of 10 minutes or more was selected to allow for sufficient data generation of interstitial glucose levels over time as recorded by CGM devices (also added to the revised MS).

  • The authors showed the results in Tables, could the authors explain and organize them in a more understandable format and use the same significant figures, in the same regard, could the authors explain better the figure captions?

We have followed the recommendations of a GEE expert for the most ideal format of data presentation. We are open to any suggestions the reviewer might have on improving the format of presentation.

  • The authors showed the results in tables, could the authors include representative plots of the glucose change in time for at least one participant in each study group?

Please see above comment. We are open to any suggestions for improvement.

  • In the main text, the authors need to verify the format of each numerical result, seems that for several results something is missing between the 2 numerical data.

Thank you for noting these typographical errors (Note: This appears to have been a combination of a typing error in a couple of cases which have now been fixed, but in a majority of cases, it appears that the ± symbol has not carried over from the word document to the pdf generated by the journal).

The authors need to verify the format of their references in the main text, some of the references are in parenthesis, and some others square brackets.

Thanks for this point...it has been corrected.

  • Could the authors include information about the influence of the temperature during their studies?

We don't have data on ambient temperature (or near body temperature) for our studies, unfortunately. Colder temperatures in general are found to decrease glycemia by ~10% in individuals with type 2 diabetes (PMID: 33764169), increase glucose uptake and improve insulin sensitivity, and CGM systems might have increased measurement error in extreme environments (see for example, PMID: 35549522). In this study, we do not expect that temperature extremes would have had an influence on glycemic control as data was recorded from sessions carried out at home, likely in temperature-controlled environments. We can allow assume that any confounder for sensor accuracy was likely equally distributed across our various comparator groups.

The authors need to organize in a better form their results.

We elected to stick with providing the results using subheadings as we felt this helps to focus the reader to the various pre-determined comparisons we elected to make.

Thank you for the opportunity to respond to your generous review! W appreciate your feedback and time...

Reviewer 4 Report

Comments and Suggestions for Authors

1. In the abstract, line 12, please define the abbreviation GEE.

2. Please ensure the values reported as mean ± SD are presented in the same format everywhere in the text. For example, line 15, either write in the same format as on line 20 or add units to both the values that are compared.

3. Section 2.2 gives an overview of the statistical analysis carried out in this study. However, the details of GEE models are still not clear. Could the authors please provide more details of the steps and parameters involved in the analysis with GEE models? Or cite references if following a protocol that was published earlier.

4. In the results and discussion sections, please check for punctuation and symbol errors. Many values are missing "±" between the mean and SD values.

5. In some results, for example line 193, the SD is larger than the mean value. Could the authors explain what it means for the data to have higher error than the mean value?

6. On line 268, kindly add references for the examples of companies.

7. A map or a chart summarizing the trends observed depending on the different conditions included in the study is suggested. This will help the reader get more clear results from this study.

Comments on the Quality of English Language

1. Line 123, please correct the sentence to - "This study was 'approved' by.."

2. Line 260, please remove the excess white space before the start of a new sentence.

Overall, please check for punctuation errors and sentence structures. The descriptions are wordy, and with many parameters and conditions in this study, the structure of descriptions used in the paper makes the information difficult to stick to the reader.

Author Response

REVIEWER 4:

  1. In the abstract, line 12, please define the abbreviation GEE.

This change has been added in.

  1. Please ensure the values reported as mean ± SD are presented in the same format everywhere in the text. For example, line 15, either write in the same format as on line 20 or add units to both the values that are compared.

We apologize for the inconsistent formatting (see reviewer 2 response above on why this may have occurred). These have been edited. Maintained as a standard format throughout (x vs y_unit). A space has also been included for consistency on either end of the ± sign for mean and SD.

  1. Section 2.2 gives an overview of the statistical analysis carried out in this study. However, the details of GEE models are still not clear. Could the authors please provide more details of the steps and parameters involved in the analysis with GEE models? Or cite references if following a protocol that was published earlier.

 Thank you for requesting this. Additional explanation on the steps followed in the GEE model and a reference to a different publication has been included.

  1. In the results and discussion sections, please check for punctuation and symbol errors. Many values are missing "±" between the mean and SD values.

The missing symbols have been added in where missing. We also noted that some of the ± symbols were in the word document but did not carry over into the pdf generated by the journal. We will further communicate with the editorial team regarding this discrepancy noted.

  1. In some results, for example line 193, the SD is larger than the mean value. Could the authors explain what it means for the data to have higher error than the mean value?

This can often happen with CGM data, since the variance in some of the various metrics like percent time below 70 mg/dL is large and sometimes skewed. Lot's of subjects had zeros' for percent time below range so this can skew the data. While we acknowledge this and report it with confidence as being accurate, we believe that this would not have any statistical bearing on the conclusions drawn as the SD values were previously accounted for in the individual means reported in the same table. The Δ values merely represent the difference between the 2 means and was analyzed to account for differences in the independent variables assessed (sex, BMI, age, exercise type and duration).

  1. On line 268, kindly add references for the examples of companies.

This information has been added into the manuscript and reference section.

  1. A map or a chart summarizing the trends observed depending on the different conditions included in the study is suggested. This will help the reader get more clear results from this study.

We have decided to put together an infographic for social media and/or the journal website, if this paper is accepted. This is typically only done in partnership with the journal.

  1. Line 123, please correct the sentence to - "This study was 'approved' by.."

This has been corrected.

  1. Line 260, please remove the excess white space before the start of a new sentence.

This has been corrected.

Overall, please check for punctuation errors and sentence structures. The descriptions are wordy, and with many parameters and conditions in this study, the structure of descriptions used in the paper makes the information difficult to stick to the reader.

Thank you for your comment (and patience)- we have proofed the revised submission. We very much appreciate your feedback and would consider any other recommendations to improve our paper. Thank you for your time and efforts to give us excellent feedback..

Round 2

Reviewer 3 Report

Comments and Suggestions for Authors

Dear Editor,

In regard to the manuscript ID- sensors-2617180 entitled “The impact of sex, body mass index, age, exercise type and exercise duration on interstitial glucose levels during exercise Short running title: CGM during exercise in healthy adults”, the authors have added and modified the requested information by the reviewers, the manuscript is more clear than previous versions and the authors have included more relevant and recent references, however before to consider the manuscript for publication at Sensors, the authors should address some points.

1)    When the authors add small sentences, in most cases a dot (.), comma (,), or spacing (  ) is missing.

2)    Could the authors verify the reference number, by adding several references, the numbering needs to be corrected in the main text. In the same regard, commas (,) are missing when adding new references in the main text.

3)    In the results section, there is a symbol, that seems the authors are editing, however, the same symbol appears in the text.

4)    In the main text, the authors have added some values with the respective units, however, the values are shown as duplicated in the provided version.

5)    The authors are showing two Figure 1 in the present manuscript version, please verify and submit only the correct one.

6)    The authors need to verify the Table numbers.

7)    The authors need to correct the format of the references, there are missing spaces or commas.

The reviewer recommends minor revision before considering the manuscript for publication.

Sincerely,

The reviewer

Author Response

Thanks sincerely for taking the extra time to review our revisions.

1) When the authors add small sentences, in most cases a dot (.), comma (,), or spacing (  ) is missing.

Thank you for noting this. We have carefully edited this final version.

2)    Could the authors verify the reference number, by adding several references, the numbering needs to be corrected in the main text. In the same regard, commas (,) are missing when adding new references in the main text.

References have been fixed.

3)    In the results section, there is a symbol, that seems the authors are editing, however, the same symbol appears in the text.

This has been fixed.

4)    In the main text, the authors have added some values with the respective units, however, the values are shown as duplicated in the provided version.

This has been located and corrected.

5)    The authors are showing two Figure 1 in the present manuscript version, please verify and submit only the correct one.

We apologize for this error. The word version has this corrected and a new higher res Tiff has been provided to the editor's office

6)    The authors need to verify the Table numbers.

Done. 

7)    The authors need to correct the format of the references, there are missing spaces or commas.

The references have been checked and corrected. Thank you for noting these errors.